# Fatigue Crack Growth Behavior and Fracture Toughness of EH36 TMCP Steel

**DOI:** 10.3390/ma14216621

**Published:** 2021-11-03

**Authors:** Qingyan Zhu, Peng Zhang, Xingdong Peng, Ling Yan, Guanglong Li

**Affiliations:** 1School of Materials and Metallurgy, University of Science and Technology Liaoning, Anshan 114051, China; huashenghunter@163.com; 2State Key Laboratory of Metal Material for Marine Equipment and Application, Anshan 114001, China; yanling_1101@126.com (L.Y.); liguanglong@ansteel.com.cn (G.L.)

**Keywords:** EH36, fatigue crack growth rate, J-integral, fracture toughness, fracture mechanism

## Abstract

The fatigue crack growth behavior and fracture toughness of EH36 thermo-mechanical control process (TMCP) steel were investigated by fatigue crack growth rate testing and fracture toughness testing at room temperature. Scanning electron microscopy was used to observe the fracture characteristics of fatigue crack propagation and fracture toughness. The results indicated that the microstructure of EH36 steel is composed of ferrite and pearlite with a small amount of texture. The Paris formula was obtained based on the experimental data, and the value of fracture toughness for EH36 steel was also calculated using the J-integral method. The observations conducted on fatigue fracture surfaces showed that there were a lot of striations, secondary cracks and tearing ridges in the fatigue crack propagation region. Additionally, there existed many dimples on the fracture surfaces of the fracture toughness specimens, which indicated that the crack was propagated through the mechanism of micro-void growth/coalescence. Based on the micromechanical model, the relationship between the micro-fracture surface morphology and the fracture toughness of EH36 steel was established.

## 1. Introduction

In recent years, many strategies have been developed for ocean vessels that have focused on maximizing their scale, minimizing their weight, and ensuring that they operate in an environmentally friendly manner and are capable of deep ocean-going. As a typical high strength and toughness ship plate steel, EH36 is mainly used in the manufacturing of large offshore platforms, large and medium-sized ocean-going ships, such as strength decks, overhead strakes or arc plates, and other key parts of hull. The properties of EH36 ship plate steel are mainly controlled by microalloying and the thermo-mechanical control process (TMCP) [1,2]. Marine ships, especially large ocean-going ships, are often used under the conditions of strong winds and waves that occur during navigation [3,4]. These conditions mean that the hull is often subject to complex stress conditions caused by the huge impact forces and periodic alternating loads that come from different directions [5,6]. In order to ensure the safety and reliability of the ocean vessels, major steel manufacturers are focused only on improving the toughness- and strength-related coupling performance of offshore structural steel, in addition to large-line heat-input welding properties and resistance to corrosion brought about by the ocean environment [7,8,9,10]. However, the fatigue resistance of high-strength offshore structural steel is also an important index that cannot be ignored. For ocean vessels, fatigue cracks are easily initiated in the local area of the hull due to their complex structures and harsh long-term working environments. When the fatigue crack growth rate reaches a certain critical value, damage to the ship structure will occur due to the propagation of fatigue cracks [11,12,13]. Barbosa et al. [14] investigated the growth rates of fatigue cracks in EH36 steel caused by submerged arc welding, and found that the threshold value of crack propagation in the fusion zone is higher than that in the base metal, because a large amount of acicular ferrite is formed in this zone, which impedes the crack propagation and can also cause the crack to deflect. Relevant papers related to the fatigue behavior mainly focus on establishing a fatigue damage model and measuring the S-N curves [15,16]. Wang et al. [17] investigated the fatigue strength of EH36 steel with T-shape welded joints at low temperature. It was found that the fatigue life at low temperature is no less than that at room temperature. However, the σ_lgN_ and CoV values of lgN at low temperature are less than those at room temperature.

With the development of fracture mechanics, the damage tolerance design is one method that has been applied to control the fatigue and fracture of large key components ocean vessels. It is generally recognized that the fatigue crack growth and fracture toughness of materials constitute the basis of damage-tolerance design [18,19]. Although much knowledge has been obtained, it is still lacking in relation to the fatigue crack growth and fracture toughness of high-strength steels. The effect of the microstructure on d*a*/d*N* shows that the propagation of cracks can be impeded by the strengthening phase in front of them. This means that the microstructure and the structure of material itself have an influence on the crack growth rate, with the sources of influence including the presence of reasonable distributions of austenite in high-strength steel and its stability during the deformation process, the high hardness of the second phase, and the incoherence of some precipitated phases in the matrix [20,21]. In addition, the crack propagation mechanism also has a great influence on fatigue crack growth rate. Wang et al. [22] investigated the fatigue resistance of EH36 weldments and found that the fatigue crack growth rate obviously decreases because of the secondary cracks and narrow fatigue striations that occur on the fatigue fracture surface. Fracture toughness is another material parameter used to characterize the ability to resist the unstable growth of cracks. As a mechanical parameter, its value can also be used as a basis for the material performance evaluation and quality assurance of typical engineering structures, including nuclear pressurized containers, oil pipelines, automotive vehicles, ship and aircraft structures and other important industrial fields [23,24,25]. The stress intensity factor *K*, the J-integral and the crack tip opening displacement (CTOD) are the most widely used parameters in the evaluation methods of fracture toughness [26,27,28]. The plane strain fracture toughness *K*_IC_ of metallic materials describes the crack propagation resistance of type-I tip cracks under slow loading in on-line elastic and negligible-yield conditions. Near the yield points of metal materials, the non-linear relationship between stress and strain can be described by the Ramberg-Osgood equation. It is especially suitable for metals that have been hardened by plastic deformation. For the high-toughness metal materials, the crack tip plastic zone radius is bigger, and thus, the linear elastic fracture mechanics cannot accurately characterize the fracture behavior. In addition, the study of material fractures from the perspective of energy has been widely recognized and applied to the evaluation methods of fracture toughness. Therefore, we need a method that contains a model of elastic and plastic fracture mechanics, such as the J-integral. This method also represents the energy required for crack propagation in elastoplastic materials. From the perspective of energy and stress field theory, the J-integral is a continuation of the *K* criterion [29]. Compared with the above parameters of fracture toughness, the J-integral method is the preferred approach the linear elastic fracture mechanics are not satisfied. Fang et al. [30] investigated the effect of crack length on the crack growth rate in the elastoplastic zone using the J-integral method, and found that the J-integral calculation method is suitable for long cracks, but it will produce errors when calculating small cracks. However, due to the complicated calculation process, basic data are lacking for the study of fracture toughness by means of the J-integral method. Therefore, it is necessary to carry out research on the fatigue behavior and fracture toughness of offshore structural steel.

As described above, EH36 ship plate steel will be subjected to alternating loads during service, which will cause fatigue damage. The rate of fatigue crack propagation will affect the service life of the EH36 steel. In the process of fracturing, the material deformation, crack initiation and propagation are accompanied by energy consumption. Fracture characteristics also differ according to the amount of energy consumed. In this paper, the fatigue crack growth rate of EH36 steel was measured at room temperature to obtain the corresponding Paris formula, and the fracture toughness value (*K*_J0.2BL_) was calculated using the J-integral method. Furthermore, the corresponding fracture mechanisms were also investigated to provide reliable theoretical support and guidance for the engineering application of ship plate steels.

## 2. Materials and Methods

In the present study, the chemical composition of EH36 high-strength steel is listed in Table 1. EH36 steel is produced by TMCP and the corresponding process parameters are as follows: the slab is heated to 1200 °C and maintained at this temperature for two hours. The initial, second and final rolling temperatures are 1050–1150 °C, 840–880° C and 810–850 °C, respectively. The optimal reduction ratio of each stage is 60% and the pass down ratio is 12%. After rolling, water cooling is carried out, the final cooling temperature is above 600 °C, and the cooling rate is 3–10 °C/s. The yield strength (R_p0.2_) and ultimate tensile strength (R_m_) of EH36 steel at room temperature are 424 MPa and 537 MPa, respectively. The elastic modulus *E* is 207 GPa, and the Poisson’s ratio ν is 0.33. The engineering stress–strain curve is shown in Figure 1. Microstructures of the etched specimen were observed with JSM 6480LV scanning electron microscopy (SEM), including the use of an energy dispersive spectrometer (EDS) and electron back-scattered diffraction (EBSD) techniques. The EBSD specimens were obtained by mechanically grinding and then electrochemical polishing the EH36 steel at a voltage of 30 V and −20 °C. The mixed solution for electrochemical polishing was composed of 5% (volume fraction) perchlorate alcohol solution.

The fatigue crack growth tests were carried out on the Instron 8801 hydraulic fatigue testing machine at ambient temperature and in a laboratory air environment, and the maximum load of the equipment was ±100 kN. The testing standard used was ISO 12108: 2012. During the experiment, a crack opening displacement (COD) gauge was installed on the specimen, and the crack length (*a*) was measured by the compliance method. The loading method was axial loading, and the stress ratio was 0.1. The test frequency was 20 Hz, and the crack propagation length was about 2 mm. The fatigue crack growth rate test was carried out on three specimens. A compact tension (CT) specimen with a size of 10 mm × 48 mm × 50 mm (*B* × *W* × *L*) was prepared for fatigue crack propagation, as shown in Figure 2a. The fracture toughness tests were carried out on the Instron 8802 hydraulic fatigue testing machine, and the maximum load of the equipment was ±250 kN. The testing standard employed was ISO 12135: 2002. Similarly, the fatigue pre-cracking process was carried out at the load ratio of R = 0.1, and the load range ΔP was 16 kN. The test frequency was 10 Hz and the pre-crack length was 3 mm. Then, the specimens were statically loaded to different crack lengths at room temperature, and the displacement rate of static loading test was 1 mm/min. Finally, the specimens were subjected to secondary fatigue until fracturing occurred. The fracture toughness test was carried out for ten specimens. The sizes of the fracture toughness specimens are also shown in Figure 2b. Before the tests, all specimens were prepared by wire cutting and with the use of a grinding machine. The difference was that the notch for the COD was located on the plane where the loading line was, and the specimen contained grooves on both sides. Furthermore, the fatigue fracture surfaces were observed by SEM. Schematic diagrams of the crack growth test and the fracture toughness test are shown in Figure 3.

## 3. Results and Discussion

### 3.1. Microstructural Observation

Figure 4 shows the microstructure of EH36 ship plate steel after deep etching, which involved ferrite and pearlite. In the corresponding SEM micrograph, it can be seen that the ferrite was mainly composed of acicular ferrite (AF) and polygonal ferrite (PF). Most of the AFs were formed in isolation in the surrounding pearlite, which can presumably be attributed to the favorability of the TMCP condition for the transformation of pearlite. Furthermore, a small amount of grain-boundary ferrite GBF can be observed along prior austenite grain boundaries, as shown in Figure 4a. In addition, the pearlite was also distributed at the ferrite grain boundaries. Figure 5 shows the distribution of the added elements, which illustrates that the added elements were evenly dissolved into the matrix.

In order to more clearly examine the microstructural features of the material, EBSD analysis was implemented, and the results are exhibited in Figure 6. The color of each grain is coded by its crystal orientation based on the [001] inverse-pole figure (IPF). The EBSD map in Figure 4 shows that the morphologies of the ferrite grains were fine and granular, and they had an average grain size of about 7.3 μm. Taking into account the φ2 value of 45° (φ1, φ = 0–90°), in addition to some important orientations in Euler space, it can be concluded that there were {112}<110> and {111}<112> textures, but that the maximum densities were not strong. According to the analysis, the material had almost no residual austenite.

### 3.2. Fatigue Properties and Fracture Toughness

During the fatigue crack growth tests, fatigue cracks in the plastic zone were first prefabricated on the specimen by the method of step-by-step load reduction. Constant load control was adopted, and the load-reduction amplitude of the adjacent level load did not exceed 20%. Then, keeping the load constant until the specimen broke, the d*a*/d*N* and Δ*K* values were obtained, and the data points that did not satisfy *W* − *a* > 4/(*K*_max_/R_p0.2_)^2^ were removed to obtain the double logarithmic curve of d*a*/d*N* − Δ*K*. Finally, the Paris formula was obtained through double logarithmic linear fitting. For EH36 ship plate steel, the increases in the crack length with the number of cycles (*N*), and the fatigue crack growth curves in terms of the Paris region obtained from the experimental record datum, are shown in Figure 7. From the variation of crack length, it can be seen intuitively that with the increasing of the number of cycles, the growth in the crack length became faster and faster. The crack-propagation curves between the crack length and the number of cycles were approximately in the form of an exponential curve, which indicates that the crack growth rate increased.

In double logarithmic coordinates, the relationship curves between d*a*/d*N* and Δ*K* are shown in Figure 8. It can be seen that there was a linear relationship between them, which satisfied the Paris formula given below: (1)$$\frac{da}{dN}=C(\Delta K^{}{)}^{m}$$
where *C* and *m* are the parameters related to the material. According to the Paris formula, larger Δ*K* values can lead to larger crack growth rates, which causes the crack growth rate to present an n power exponential form with the propagation of the crack. According to the datum, the individual and average Paris formulas for three samples and the corresponding values of *C* and *m* can be obtained using the least square linear fitting method, as shown the solid red line in Figure 8. The corresponding values of *C* and *m* are also listed in Figure 8.

In this paper, the fracture toughness of EH36 steel was measured using the J-integral (J) versus crack propagation quantity (Δ*a*) resistance curve at room temperature. During the fracture toughness tests, different specimens were statically loaded to different crack growth lengths, and then unloaded. Meanwhile, the load and displacement data of the loading line were collected and recorded. In order to accurately measure the length of crack growth, all specimens needed to be thermally colored. The heating temperature was 350 °C, and the heating time was 30 min. Then, the fatigue crack length was measured and the corresponding J-integral value was calculated according to the relevant formula. The mathematical expression is shown as follows:(2)$$J=\alpha +\beta (\Delta a{)}^{\gamma }$$
where *α*, *β* and *γ* are the parameters related to the material. According to the fatigue crack fracture, nine crack lengths of the prefabricated crack front and nine crack lengths of the crack propagation front were measured using an optical microscope. Using the average length of the prefabricated cracks (*a*_0_) and of the propagation cracks (*a*), the value of Δ*a* was calculated using Equation (3).
(3)$$\Delta a=\frac{1}{8}(\frac{a_{1}+a_{9}}{2}+\sum_{i=2}^{8}a_{i})-\frac{1}{8}(\frac{a_{01}+a_{09}}{2}+\sum_{i=2}^{8}a_{0i})$$

Furthermore, the value of J was calculated using the loading and unloading curves. The loading and unloading curves of different fracture toughness specimens are shown in Figure 9.

Using the Δ*a* as the abscissa and J as the ordinate, the J-Δ*a* diagram is shown in Figure 10. There is a passivation line in the figure, as shown by tilted solid black line, the expression of which is J = 3.75R_m_∆*a* = 2014∆*a*. Drawing a parallel line of passivation line through (0.1, 0) and taking the region to the right of parallel line as the effective region, all data points within the effective region were fitted according to Equation (2). A new J-Δ*a* resistance curve could be obtained, as shown by the solid red line in the figure. Then, Equation (2) could be rewritten as J = 1085∆*a*^0.376^ (α = 0, β = 1085, γ = 0.376). At this point, the parallel line of the passivation line was drawn through at the point (0.2, 0) to obtain the y-coordinate of the intersection point of the parallel line and the J-Δ*a* resistance curve, which represented the fracture toughness of the EH36 steel. The value of J_0.2BL_ was 926 kJ/m^2^. *K*_J0.2BL_ could be calculated using Equation (4) as follows:(4)$$J=\frac{1-\nu^{2}}{E}K^{2}$$

Finally, the value of *K*_J0.2BL_ was 464 MPa·m^1/2^.

### 3.3. Morphological Features of Fracture Surface

In order to analyze the fatigue fracture mechanisms of the EH36 steel, the fatigue fracture and crack morphology were observed and characterized at the micro level. As is well known, the fatigue fractography consists of several different zones, including the initiation, propagation and final fracture of the fatigue crack. Figure 11 displays the fracture morphology of the propagation region for three fatigue crack propagation specimens at different magnifications. The direction of fatigue crack propagation is indicated by a long white arrow. Figure 11b,d,f are the enlarged views of the white box region in Figure 11a,c,e. As can be seen from the diagram, the typical characteristics of striations could be observed in the crack-expanding region. It is theoretically possible that every fatigue cycle would produce a fatigue striation in the Paris region. The average striation distance can reflect the value of the crack growth rate. That is to say, the narrower the fatigue striation distance, the slower the growth of the fatigue crack, which indicates a better resistance to fatigue propagation [31,32]. In addition to fatigue striations, secondary cracks and tearing ridges could also be observed in this zone. This indicated that there was a large stress concentration at the fatigue crack tip, and that the crack propagation was prone to deflection under normal stress, resulting in a relatively rough fatigue crack propagation zone. Extensive studies have shown that the fatigue limit of ordinary carbon steel will increase with the decreasing of the subgrain size [33]. Heat treatment can effectively change the microstructure of eutectoid steel, that is, it can change the spacing of the thin ferrite and pearlite sections in the steel, and thus, change the fatigue property of the material. Furthermore, the presence of partial {112}<110> and {111}<112> textures in the microstructure of EH36 steel will lead to the anisotropy of polycrystalline alloy, which will, to a certain extent, influence the deformation behavior [34]. At the fatigue crack tip, due to the large stress concentration, the stress near the crack tip will exceed the yield stress of the material and plastic deformation will occur, influencing the size of the plastic zone. The size and distribution of the plastic zone will affect the fatigue crack propagation behavior of the material. The existence of the texture and the plastic zone may cause the crack to deflect at the micro scale and form a secondary crack, which will reduce the driving force of the crack tip growth, indirectly hindering the forward propagation of the crack, and this is beneficial in terms of improving the fatigue resistance of the material.

As an important part of fracture toughness research, the observation of fractures is essential due to the fracture surface preserving traces of the damage to the specimen that occurs during loading, which will help in understanding the fracture properties and deformation behavior of the material. The fracture morphologies of ductile fatigue toughness specimens were observed at the macro and the micro level, and the results are shown in Figure 12. Since the specimens were colored after loading and unloading in the fracture toughness test, the regions corresponding to different stages could be clearly distinguished from the crack surfaces, as shown in Figure 12a,b. It can be seen that the boundary lines of several regions on the fracture surface were basically parallel. From right to left, the fracture surface presents the fatigue pre-crack zone, the static loading process zone, the second fatigue zone and the final instability fracture zone, as shown by the red horizontal arrow in Figure 12a. The final fracture zone and rapid crack growth zone were relatively rough, but the final instability fracture zone exhibited the phenomenon of necking. The transition stage from the second fatigue region to the final instability fracture region is marked as Ⅰ. The edge between the static loading process zone and the second fatigue zone is flagged as Ⅱ. The partial amplification of the red box area in Figure 12a offers a clearer representation of the four types of regions and edges, as shown in Figure 12b. The fractographs of the secondary fatigue zone are shown in Figure 12c,d, and Figure 12f,g show the local enlarged view of Figure 12c,d, respectively. Similarly to the crack propagation specimens, there are fatigue striations, tearing ridges and secondary cracks in the secondary fatigue zone. At the initial stage of this region, due to the relatively low stress intensity factor, the size of plastic zone at the crack tip was small, resulting in the striations being relatively shallower and the crack growth zone being relatively flat. With the increase in the number of cycles, the stress intensity factor increased and the crack propagation accelerated, leading to clearer fatigue striations and an increase in secondary cracks. Combined with the fracture morphology of both tests, it can be inferred that the fracture surfaces of EH36 steel exhibit a typical transcrystalline fracture type.

In addition to the typical fatigue fracture characteristics, there were also a lot of ductile fracture characteristics in the static loading process zone on the fracture surface. For instance, many dimples of different sizes could be observed, as shown in Figure 12e,h. As is widely known, the growth/coalescence and the cleavage model of micro-voids are two main micro fracture mechanisms that occur during the crack propagation of materials [35,36]. Stress concentration occurs at the crack tip due to the occurrence of plastic deformation during loading. If the local stress at the crack tip first reaches the cleavage fracture strength of the grain, the main crack will propagate forward through the cleavage. At this time, the fracture surface is relatively smooth. On the contrary, if the local stress at the crack tip does not reach the cleavage fracture strength of the grain, a large amount of dislocation pile-up will be generated near the crack tip due to stress concentration, and the micro-voids will nucleate in dislocation pile-up mode. Then, under the action of plastic strain, the micro-voids are promoted to grow via a dislocation movement around the micro-voids, polymerize with other micro-voids, and finally form dimples [37]. At this time, the main crack is connected to the dimple in front of it to form a new main crack, thus realizing the crack propagation behavior. The above two microscopic mechanisms of crack propagation generally exist simultaneously on the surface of fatigue fractures depending on the competitive relationship between stress and strain at the crack tip. Therefore, different materials will have different fracture morphologies during crack propagation, and the EH36 steel studied in this paper formed dimples with different sizes under static loading, which is a typical micro-void growth/coalescence mechanism.

### 3.4. Analysis of Fracture Morphology and Fracture Toughness

Different fracture propagation mechanisms indicated that the propagation resistance and energy consumption were not consistent during fatigue crack propagation, and the corresponding fracture toughness was also different. In microstructures, this will lead to the formation of different fracture morphologies. Therefore, the fracture toughness could be characterized by the microscopic fracture morphologies of the material. Based on the micromechanical model [38], the commonly used quantitative relationship between fracture toughness (*K_C_*) and dimple size is as follows:(5)$$K_{C}=(E\cdot R_{m}\cdot h{)}^{1/2}$$
where *h* is the dimple height. To facilitate measurement, the diameter (*d*) can be used instead of the height of dimples for calculation.

However, for dimples of different sizes, the ratio of height to diameter is often inconsistent. Therefore, based on the previous studies and the fracture morphology of EH36 steel, the dimples were divided into large dimples and small dimples. For large dimples (*d* > 10 µm), the height–diameter ratio could be considered to be about 1, that is, *d_l_* = *h_l_*, where *h_l_* and *d_l_* are the height and diameter of the large dimples, respectively. For the small dimples (*d* < 10 µm), the height–diameter ratio detected and verified by FIB technology was about 0.5, that is, *d_s_* = 2*h_s_*, where *h_s_*, *d_s_* are the height and diameter of the small dimples, respectively. The dimple-free region can be considered as a cleavage plane, and its formation energy is far less than that of dimples. Accordingly, it could be considered as a special dimple with the height of 0. The fracture toughness of material could be derived from the volume fraction of dimples of different sizes, and Equation (5) could be converted to:(6)$$K_{C}=D\cdot \left[\sum S_{l}\cdot (E\cdot R_{m}\cdot d_{l}{)}^{1/2}+\sum S_{s}\cdot (\frac{1}{2}E\cdot R_{m}\cdot d_{s}{)}^{1/2}\right]$$
where *S_l_* and *S_s_* are the area fraction of large and small dimples, respectively. *D* = 11 is a coefficient related to the width to thickness ratio of specimen. In this way, the quantitative relationship between the micro-fracture characteristics and fracture toughness of EH36 steel was obtained.

According to the fracture morphology in Figure 12, the diameter and number of dimples were counted by SEM, and then the percentages of area occupied by dimples with different sizes were analyzed, as shown in Figure 13. Although the average diameter of the dimples was 11.12 µm, due to their relatively small size, small dimples accounted for only 16.69% of total area. The fracture toughness corresponding to all dimples could be calculated using Equation (6). As the number of statistical dimples increased, the value of fracture toughness gradually increased, as shown in the red curve in Figure 13. The highest point of the curve was the fracture toughness of EH36 steel calculated using dimples. As mentioned above, the J-integral is also a method that is used to calculate fracture toughness from the perspective of energy. The value of *K*_J0.2BL_ is also shown in the dashed line in Figure 11; it can be seen that *K*_J0.2BL_ was slightly higher than *K_C_*. In general, for highly ductile materials, the fracture toughness of the material is the sum of the energy required for plastic deformation, micro-crack initiation/propagation and micro-void growth/ coalescence [39]. This small difference may have been caused by the fact that when calculating the value of *K_C_* using Equation (6), only the energy absorption under the ductile fracture formed by plastic deformation was considered, while the energy consumed to form the local tiny brittle cleavage plane was ignored. Furthermore, Frómeta [40] also gave similar results, believing that the length of the crack at the initial stage of crack propagation and at the later stage of crack propagation would have a certain influence on the calculated value, causing the value calculated using the dimples in the static loading process zone at the front of crack to be lower than that calculated using the J-integral.

Through the above analysis, we linked fracture toughness with fracture surface morphology, and fracture morphology was attributed to different fracture mechanisms. From the perspective of energy, there are differences in energy consumption among different fracture mechanisms. It can be considered that the greater the amount of energy consumed during fracture, the larger the fracture toughness of the material. The relationship between the fracture mechanism, the micro-fracture surface morphology and the fracture toughness in the material is shown in Figure 14. For EH36 steel, under the condition of pre-cracking, the more numerous and larger the dimples on the fracture surface—representing the characteristics of ductile fracture—are, the higher the value of fracture toughness (*K_C_*) will be. Furthermore, compared with small dimples, the formation of large dimples will consume more energy, thus improving the fracture toughness of EH36 steel. In this paper, by analyzing the fracture mechanism of EH36 steel, the influence of the cleavage plane, small dimples and large dimples on the fracture toughness is discussed, which can provide a reasonable theoretical basis for safe application in engineering.

## 4. Conclusions

The parameters of crack propagation were obtained and the value of K_J0.2BL_ was calculated. The crack propagation behavior and fatigue fracture mechanism of EH36 steel were investigated at room temperature. The corresponding conclusions can be summarized as follows:

(1)The microstructure of EH36 steel was composed of fine ferrite and pearlite, with an average grain size of about 7.3 μm. The Paris formula was obtained by means of linear fitting, and the corresponding average values of C and m were equal to 1.975 × 10^−9^ and 3.327, respectively. The fracture toughness of the EH36 steel was also calculated using the J-integral method, and the value of K_J0.2BL_ was 464 MPa·m^1/2^.(2)The fatigue fracture surfaces of the EH36 steel exhibited typical transcrystalline fracture characteristics, accompanied by fatigue striations, secondary cracks and tearing ridges. There were many dimples with different sizes in the static loading process zone of the fracture toughness tests, which indicated that EH36 steel has good toughness and presents the characteristics of ductile fracture under static loading conditions.(3)Based on the energy consumption analysis of the fracture morphology, the relationship between the micro-fracture surface morphology and the fracture toughness of EH36 steel was established, and the fracture toughness obtained was close to that calculated using the J-integral method.

## Figures and Tables

**Figure 1 materials-14-06621-f001:**
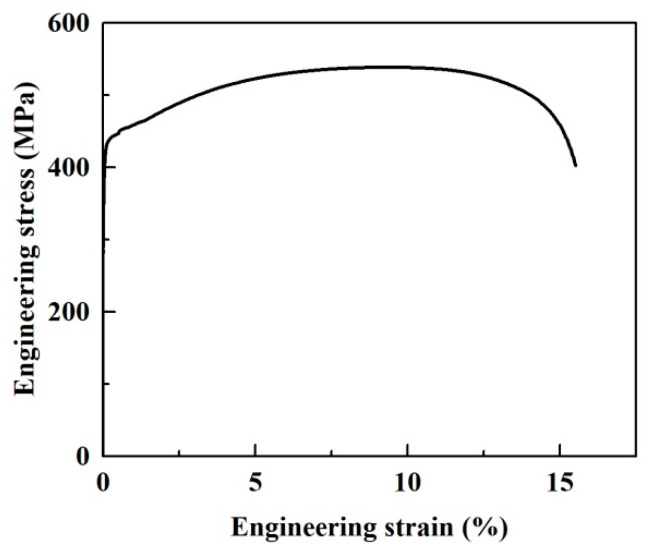
Engineering stress-strain curve of EH36 steel.

**Figure 2 materials-14-06621-f002:**
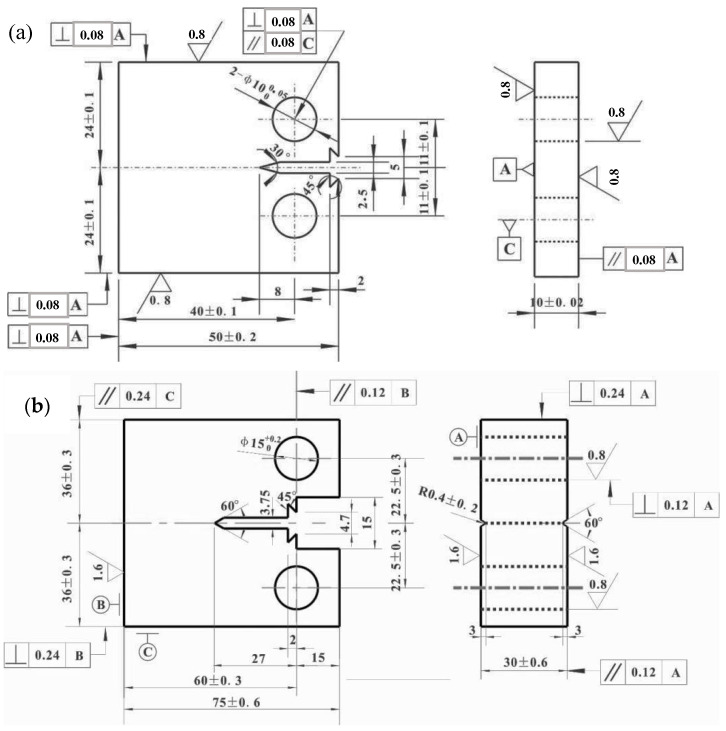
Size of compact tension specimens (Unit: mm): (**a**) specimen used for fatigue crack growth rate test; (**b**) specimen used for fracture toughness test.

**Figure 3 materials-14-06621-f003:**
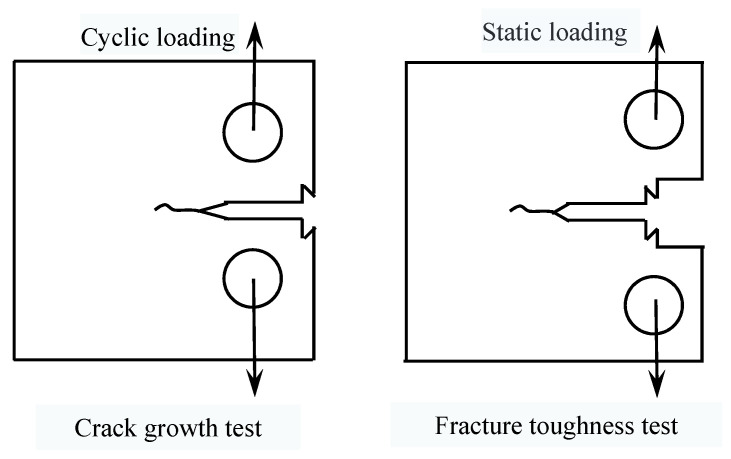
Schematic diagrams of crack growth test and fracture toughness test.

**Figure 4 materials-14-06621-f004:**
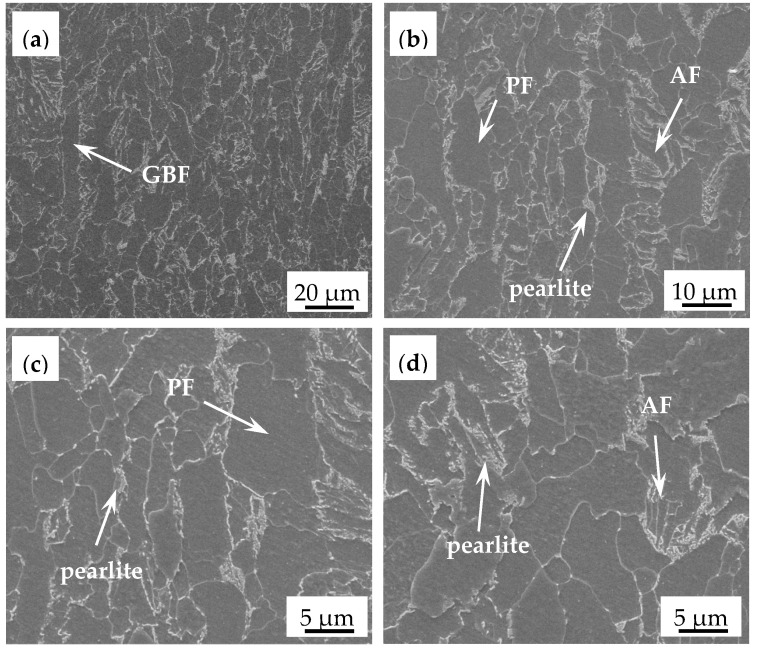
SEM micrographs of EH36 steel: (**a**) ×2000; (**b**) ×4000; (**c**) ×8000; (**d**) ×8000.

**Figure 5 materials-14-06621-f005:**
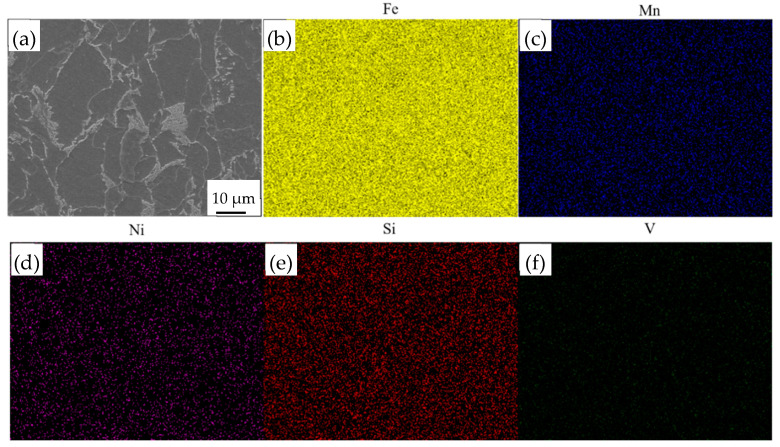
Distribution of elements in EH36 steel: (**a**) SEM, ×4000; (**b**) Fe; (**c**) Mn; (**d**) Ni; (**e**) Si; (**f**) V.

**Figure 6 materials-14-06621-f006:**
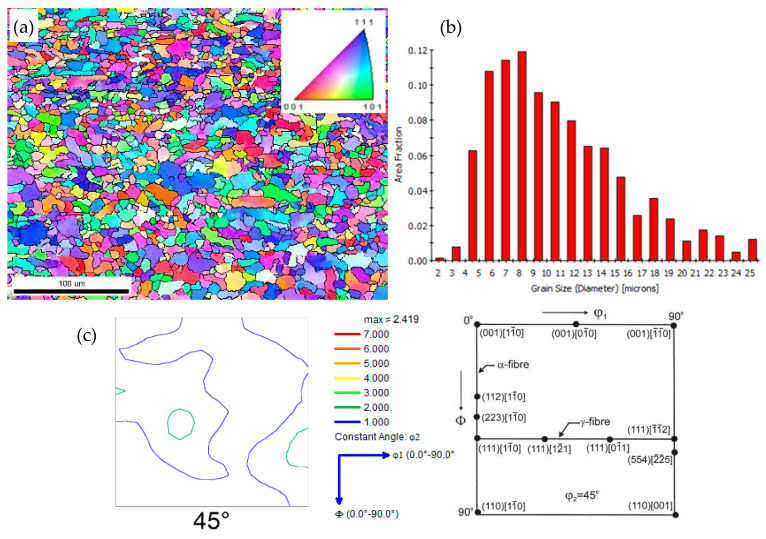
EBSD observation of EH36 steel: (**a**) EBSD map; (**b**) grain size; (**c**) corresponding textures.

**Figure 7 materials-14-06621-f007:**
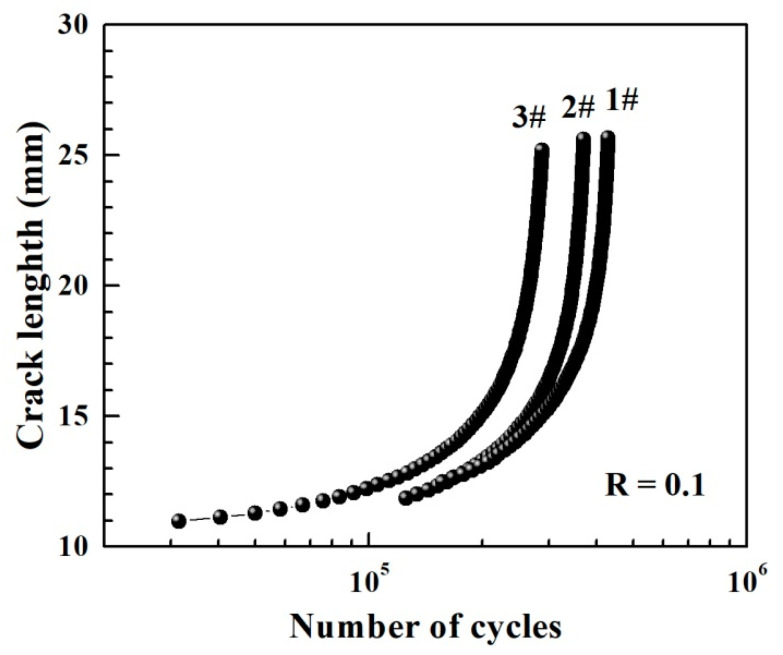
Relationships between fatigue crack length and number of cycles for 1#–3# specimens of EH36 steel.

**Figure 8 materials-14-06621-f008:**
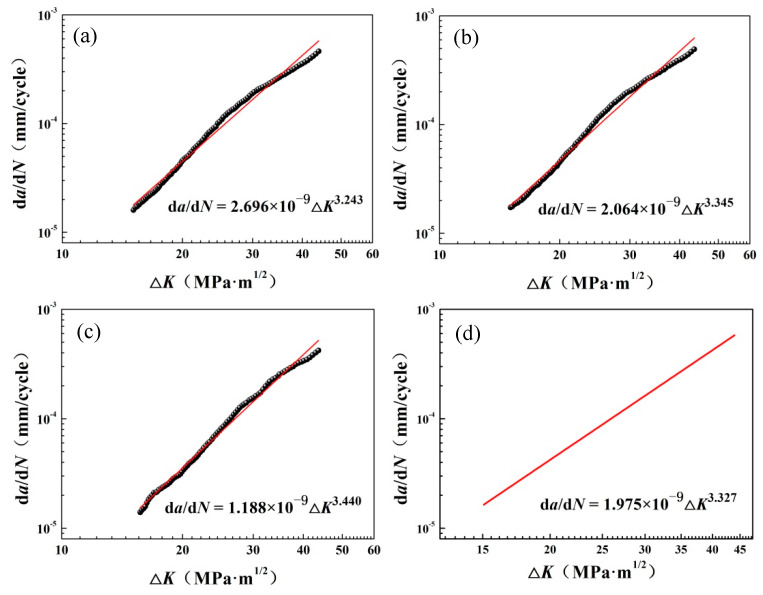
Fatigue crack propagation rate as a function of stress intensity factor range for EH36 steel: (**a**) 1# specimen; (**b**) 2# specimen; (**c**) 3# specimen; (**d**) average value of three specimens.

**Figure 9 materials-14-06621-f009:**
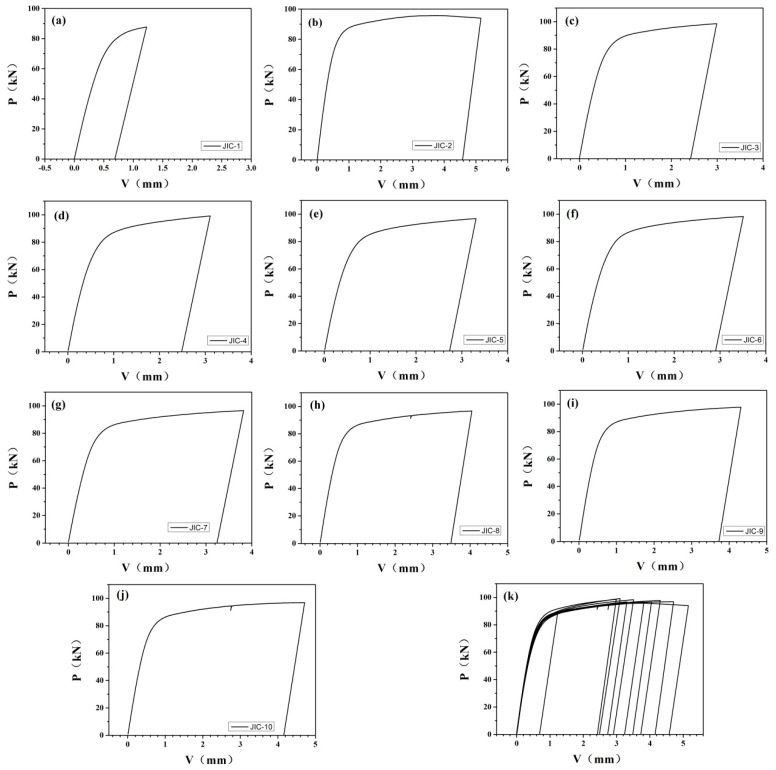
Force versus notch opening displacement curves of EH36 steel for fracture toughness specimens: (**a**) 1# specimen; (**b**) 2# specimen; (**c**) 3# specimen; (**d**) 4# specimen; (**e**) 5# specimen; (**f**) 6# specimen; (**g**) 7# specimen; (**h**) 8# specimen; (**i**) 9# specimen; (**j**) 10# specimen; (**k**) 1#−10# specimens.

**Figure 10 materials-14-06621-f010:**
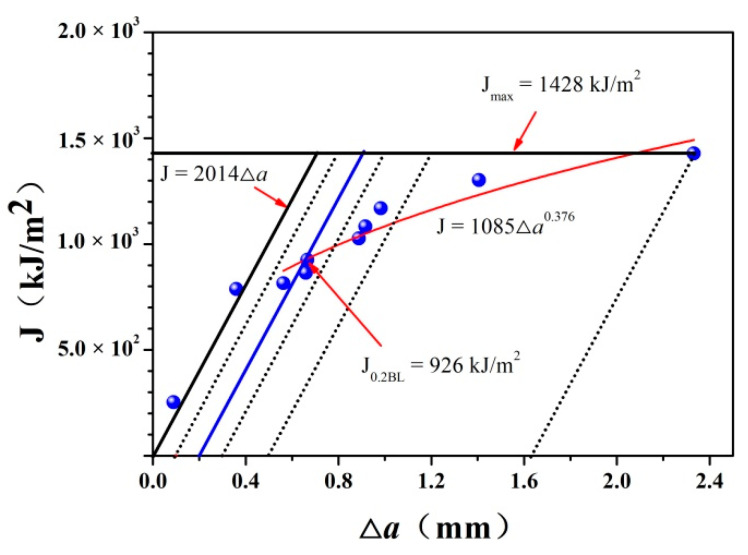
Fracture toughness J-∆*a* resistance curve of EH36 steel.

**Figure 11 materials-14-06621-f011:**
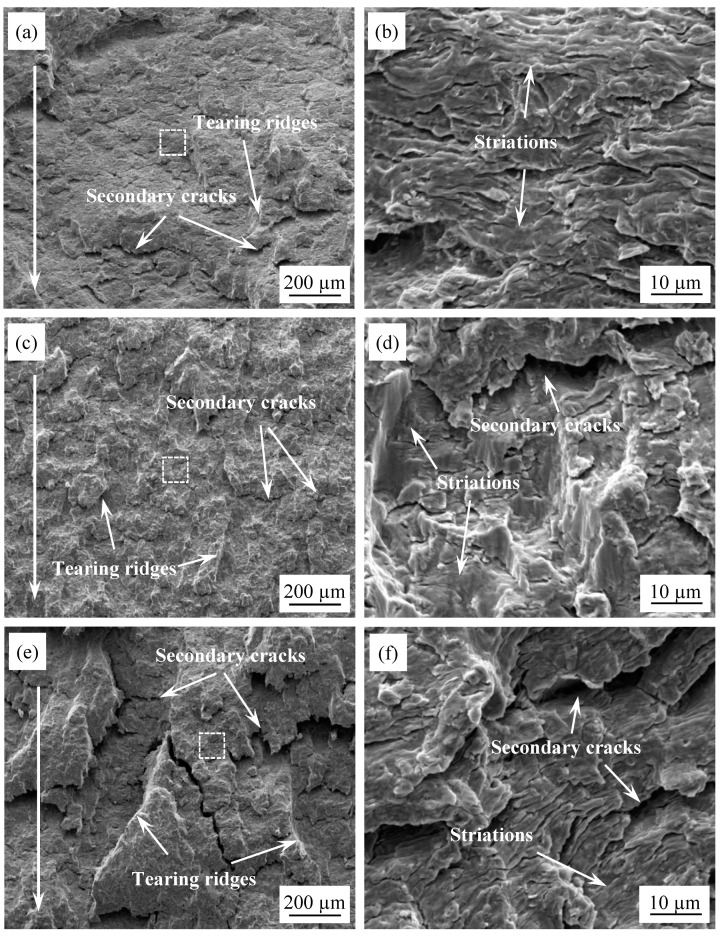
Fractographs of fatigue crack propagation region: (**a**) 1# specimen ×200; (**b**) 1# specimen, ×4000; (**c**) 2# specimen, ×200; (**d**) 2# specimen, ×4000; (**e**) 3# specimen, ×200; (**f**) 3# specimen, ×4000.

**Figure 12 materials-14-06621-f012:**
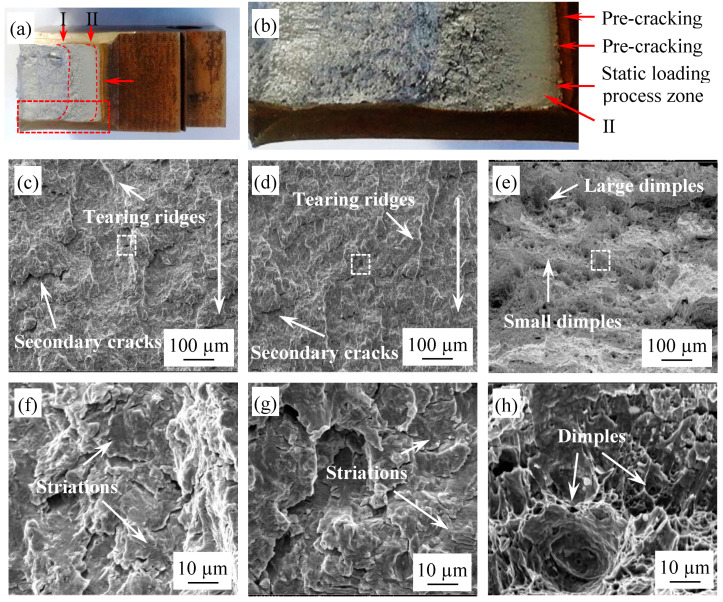
Fractographs of ductile fatigue toughness specimens: (**a**,**b**) macro fracture morphology; (**c**–**h**) micro fracture morphology.

**Figure 13 materials-14-06621-f013:**
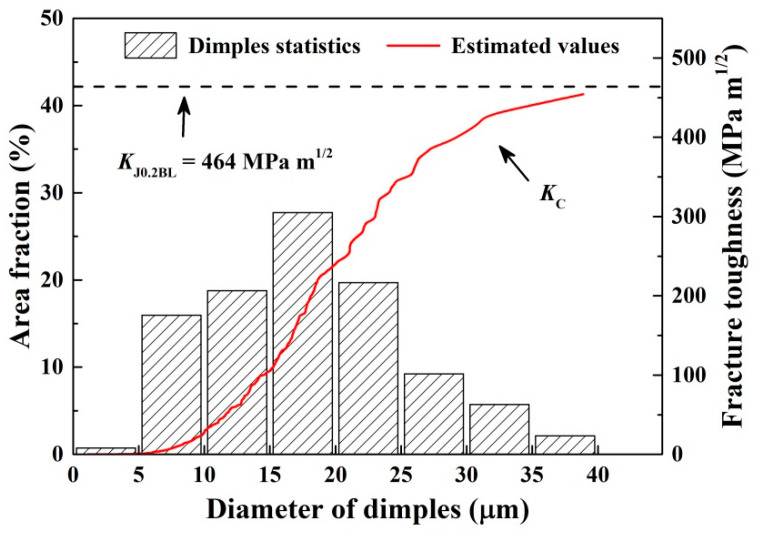
Dimple size distributions and corresponding fracture toughness of EH36 steel.

**Figure 14 materials-14-06621-f014:**
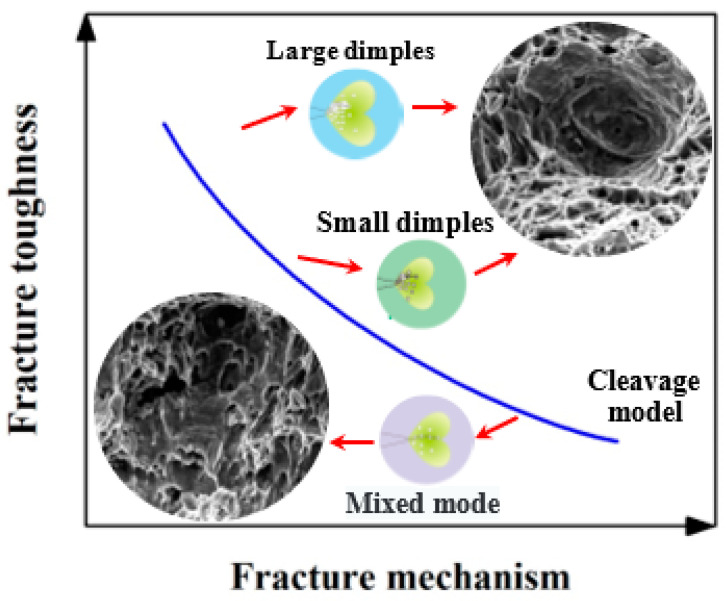
Relationship between fracture mechanism, micro-fracture surface morphology and fracture toughness.

**Table 1 materials-14-06621-t001:** Chemical composition of EH36 steel (mass%).

C	Si	Mn	Ni	V	Nb	Ti	P	S	Fe
0.10	0.15	1.55	0.25	0.05	0.04	0.012	0.002	0.003	Bal.

## Data Availability

Data are available in a publicly accessible repository.

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
