# Peer review of "Fatigue Crack Growth Behavior and Fracture Toughness of EH36 TMCP Steel"

_materials, 2021, doi:10.3390/ma14216621_

Round 1
Reviewer 1 Report
More details should be added in Section 2:
- There is no information on how the specimens were fabricated. In particular, it should be described how the notches were cut for the two CT specimen types.
- There is no reference to any standard concerning CT test. It must be added.
- It should be explained how the specimens’ dimensions were determined. Is there a reason why dimensions of CT specimen used for fatigue crack growth rate test and CT specimen used for fracture toughness test are different? – e.g. hole diameters as well as specimen height and length are different. As it can be seen, width of specimen in Figure 1b is equal to 30mm (which is a half of 60mm – this is in a good agreement with ASTM5045) whereas width of specimen shown in Figure 1a is equal to 10mm (which is a quarter of 40mm). This should be commented.
- Is this really important for the test to obtain accurate values of surface roughness parameters? – e.g. roughness of a back surface of specimen shown in Figure 1b is equal to 1.6.
- What kind of clevis was used? Was it a special one?
- What was the velocity of testing machine crosshead in case of both tests? Was it 1mm/min? It should be given.
- Equipment in testing conditions can be additionally presented for both tests (schematic or real view) to make it more clear for a reader.
- Moreover, a resolution of the images in Figure 1 seems to be too low.
- Move Fig. 12 out of Conclusions
Author Response
Dear Editors and Reviewer,
We would like to express our great appreciation to your comments on our paper. We have studied comments carefully and have made correction which we hope meet with approval. Revised portion are marked in red in the paper.
- There is no information on how the specimens were fabricated. In particular, it should be described how the notches were cut for the two CT specimen types.
Response: This sentence “Before the tests, all samples were prepared by wire cutting and grinding machine.” was added in Section 2.
- There is no reference to any standard concerning CT test. It must be added.
Response: The tests standards were add in Section 2. ISO 12108: 2012 and ISO 12135: 2002.
- It should be explained how the specimens’ dimensions were determined. Is there a reason why dimensions of CT specimen used for fatigue crack growth rate test and CT specimen used for fracture toughness test are different? – e.g. hole diameters as well as specimen height and length are different. As it can be seen, width of specimen in Figure 1b is equal to 30mm (which is a half of 60mm – this is in a good agreement with ASTM5045) whereas width of specimen shown in Figure 1a is equal to 10mm (which is a quarter of 40mm). This should be commented.
Response: The minimum recommended thickness was generally used according to both test standards.
- Is this really important for the test to obtain accurate values of surface roughness parameters? – e.g. roughness of a back surface of specimen shown in Figure 1b is equal to 1.6.
Response: Values of surface roughness parameters were based on the test standard.
- What kind of clevis was used? Was it a special one?
Response: Use conventional flat bottomed clevis according to test standard.
- What was the velocity of testing machine crosshead in case of both tests? Was it 1mm/min? It should be given.
Response: The fatigue tests were stress-controlled with a stress ratio of 0.1 and a frequency of 10/20 Hz. In order to ensure deformation in the elastic region, the displacement rate of static loading test is 1 mm/min. Test parameters were added in Section 2
- Equipment in testing conditions can be additionally presented for both tests (schematic or real view) to make it more clear for a reader.
Response: Schematic diagrams of crack growth test and fracture toughness test were added in Section 2.
- Moreover, a resolution of the images in Figure 1 seems to be too low.
Response: The resolution of Figure 1 has been adjusted.
- Move Fig. 12 out of Conclusions
Response: Fig. 12 has been moved from Conclusions.
We appreciate for Editors/Reviewers’ warm work earnestly, and hope that the correction will meet with approval.
Once again, thank you very much for your comments and suggestions.
Reviewer 2 Report
I am asking the authors to anticipate the nomenclature at the beginning of the paper, before the introduction. Please include all abbreviations, symbols and markings in it. This will make the manuscript easier to read and understand.
Instead of the words "sample" or "samples", please use the words "specimen", "specimens".
Correct abstract, correct introduction with a short but factual review of the literature.
Properly described material, experiment and geometry of the specimens used, supplemented with tables with chemical composition and a drawing of the specimen geometry. Please add only information about the material hardening degree in terms of the exponent and the constant in the Ramberg-Osgood law.
The paper contains a proper description of the microstructure - I have no comments to it.
The authors properly present the experimental results obtained in the field of fatigue tests - it is a pity that there are not yet three or five specimens here. The approach leading to the determination of fracture toughness - appropriate method, should be considered correct, especially since the material was quite brittle.
In some cases, the correct use of "The" and "a" is missing. The authors unnecessarily combine the description of the patterns with the rest of the text in the manuscript - it is worth changing this - the paper will be clearer.
Overall, I rate the manuscript very well. I suggest a minor revision.
Author Response
Dear Editors and Reviewer,
We would like to express our great appreciation to your comments on our paper. We have studied comments carefully and have made correction which we hope meet with approval. Revised portion are marked in red in the paper.
1. I am asking the authors to anticipate the nomenclature at the beginning of the paper, before the introduction. Please include all abbreviations, symbols and markings in it. This will make the manuscript easier to read and understand.
Response:All abbreviations, symbols and markings are added before the introduction.
2. Instead of the words "sample" or "samples", please use the words "specimen", "specimens".
Response: The words "specimen" and "specimens" are used.
3. Please add only information about the material hardening degree in terms of the exponent and the constant in the Ramberg-Osgood law.
Response:The information of material hardening degree in terms of the exponent and the constant in the Ramberg-Osgood law was added in Section 1.
4. In some cases, the correct use of "The" and "a" is missing. The authors unnecessarily combine the description of the patterns with the rest of the text in the manuscript - it is worth changing this - the paper will be clearer.
Response:Part of “the" and "a" were modified.
We appreciate for Editors/Reviewers’ warm work earnestly, and hope that the correction will meet with approval.
Once again, thank you very much for your comments and suggestions.
Reviewer 3 Report
Dear Authors,
Thank you for the opportunity to read and review your article.
The topic is timely, interesting and necessary.
The overall layout is good , but some of the figures and tables are not very readable and the font is too small - this reduces the quality of the article.
The title of the article is fine.
The abstract is clear, concise and absolutely ideal.
The introductory paragraph and the overall justification for the EH36 research is journal and general.
The reader finds the chemical composition, all mechanical properties and also a detailed description of fatigue testing. The choice of tests corresponds to current trends and practices.
The results, discussion and description is extensive, detailed and I do not see a problem here.
The analysis of the microscopic images is again in line with the present.
The paper introduces the material, but the conclusions are a bit vague - the advantages of steel and its use need to be added.
Regards,
Translated with www.DeepL.com/Translator (free version)
Author Response
Dear Editors and Reviewer,
We would like to express our great appreciation to your comments on our paper. We have studied comments carefully and have made correction which we hope meet with approval. Revised portion are marked in red in the paper.
1. The overall layout is good , but some of the figures and tables are not very readable and the font is too small - this reduces the quality of the article.
Response:Increased the size of figure 1 to make the font easier to see.
2. The paper introduces the material, but the conclusions are a bit vague - the advantages of steel and its use need to be added.
Response:Conclusion 2 was slightly modified.
We appreciate for Editors/Reviewers’ warm work earnestly, and hope that the correction will meet with approval.
Once again, thank you very much for your comments and suggestions.
Reviewer 4 Report
Recently, many researchers have again turned to research on low-carbon low-alloy steels, which is associated with the expansion of their application as a structural material. The authors have studied the fracture processes in fatigue crack propagation testing as well as the fracture toughness characteristics of low-carbon low-alloy steel specimens. The advantage of this study is the calculation of the fracture toughness parameter by two methods, namely, using the J-integral and using the energy consumption analysis; the results showed good agreement between the values. Therefore, the work is relevant and interesting. However during the careful review of the submitted manuscript some issues arose. Therefore a moderate revision is required before the manuscript can be published. The issues and comments as listed below:
1) lines 50-51. What is ‘σlgN’ and ‘CoV of lgN’? It should be clarified.
2) line 73. KIC - subscript needed
3) lines 102-103. Can you provide information on the initial processing of steel?
4) line 115. Is ‘COD’ a Crack Opening Displacement? A decryption should be given at the first mention.
5) lines 288-308. The authors discuss the mechanisms of the fracture process in relation to the studied steel. However, it is not clear from the description of the results - in which fracture regions the authors observed the dimple fracture. Was it all over the fracture surface or only in the final fracture zone? Looking at the fractographic images of Figure 10, one can see extended planes of brittle or quasi-brittle fracture. Apparently, both mechanisms of fracture take place in this case. So, the authors should discuss this point in more detail. In this regard, Conclusion 2 may require correction.
6) lines 345-352. The difference in fracture toughness values ​​determined by the two methods seems to be very small. Therefore, you should not focus on it. In any case, there is a measurement error in determining the fracture toughness by each of these methods.
7) Figure 5 shows three curves for three different samples, right? It is necessary to indicate this in the figure caption or on the figure itself. By the way, were these three samples different? Or did you mean that there are three identical specimens of the same steel, but not three different samples?
8) Only approximating line should be in Figure 6d. Other experimental curves are already in Figure 6 a, b, and c.
9) Figure 7 shows the curves for 9 specimens, but the Method part reports 10 specimens.
Author Response
Dear Editors and Reviewer,
We would like to express our great appreciation to your comments on our paper. We have studied comments carefully and have made correction which we hope meet with approval. Revised portion are marked in red in the paper.
1) lines 50-51. What is ‘σlgN’ and ‘CoV of lgN’? It should be clarified.
Response:In the literature [17], lgN is the log of the number of cycles (N), can be used as the tested fatigue life. σlgN is the standard deviation of the tested fatigue life. CoV is defined as the ratio of the standard deviation to the mean value.
2) line 73. KIC - subscript needed
Response:The error was modified.
3) lines 102-103. Can you provide information on the initial processing of steel?
Response:Production process of EH36 steel were added in Section 2.
4) line 115. Is ‘COD’ a Crack Opening Displacement? A decryption should be given at the first mention.
Response:Yes, a decryption was given at the first mention in Section 2.
5) lines 288-308. The authors discuss the mechanisms of the fracture process in relation to the studied steel. However, it is not clear from the description of the results - in which fracture regions the authors observed the dimple fracture. Was it all over the fracture surface or only in the final fracture zone? Looking at the fractographic images of Figure 10, one can see extended planes of brittle or quasi-brittle fracture. Apparently, both mechanisms of fracture take place in this case. So, the authors should discuss this point in more detail. In this regard, Conclusion 2 may require correction.
Response:Dimples are observed in the static loading process zone, which corresponds to ductile fracture under static stretching. Now, the arrows indicate some large and small dimples in Figure 12(e) and 12(h). Conclusion 2 was slightly modified.
6) lines 345-352. The difference in fracture toughness values ​​determined by the two methods seems to be very small. Therefore, you should not focus on it. In any case, there is a measurement error in determining the fracture toughness by each of these methods.
Response:Yes, there are differences between different methods due to different principles.
7) Figure 5 shows three curves for three different samples, right? It is necessary to indicate this in the figure caption or on the figure itself. By the way, were these three samples different? Or did you mean that there are three identical specimens of the same steel, but not three different samples?
Response:Now, FIG. 7 shows three groups of repeated tests under the same conditions in the fatigue crack propagation test. The value of calculated average is more reliable. The using of three test samples is also indicated in Section 2.
8) Only approximating line should be in Figure 6d. Other experimental curves are already in Figure 6 a, b, and c.
Response:Now, experimental curves were deleted and approximating line was retained in Figure 8d.
9) Figure 7 shows the curves for 9 specimens, but the Method part reports 10 specimens.
Response:Now, in order to make the picture format neat, 9 curves were placed before. Now, all 10 curves are placed in Fig.9
We appreciate for Editors/Reviewers’ warm work earnestly, and hope that the correction will meet with approval.
Once again, thank you very much for your comments and suggestions.
Reviewer 5 Report
The manuscript can be published after minor revision.
1-Some sentences are difficult for the reader to understand. Therefore, the final version of the article needs to be edited.
2- It should be checked grammatically. For example, the parallel structure is not observed in the text.
3- The chemical composition of the raw material and mechanical properties such as yield stress, ultimate tensile strength, etc. are the results of tests performed by the authors? if yes, please describe the tests and report the results data such as the stress-strain diagram. if not, please refer to an appropriate reference.
4- How did you create the initial crack in various samples? Also, explain the author's method of ensuring the exact size of the pre-crack in all samples.
5- During the tests, what is the equipment used to measure cracks at different stages. please describe your method in full detail?
6- Related to Figure 5, what is the measurement accuracy of crack length?
7- Related to Figure 7, the authors stated in the text that 10 samples were used to perform the test, But in this figure, there are nine diagrams as the test results. In addition, it is strongly suggested to re-draw all diagrams in one image for comparison easier than ago.
Author Response
Dear Editors and Reviewer,
We would like to express our great appreciation to your comments on our paper. We have studied comments carefully and have made correction which we hope meet with approval. Revised portion are marked in red in the paper.
1-Some sentences are difficult for the reader to understand. Therefore, the final version of the article needs to be edited.
Response:Some sentences are modified and highlighted in red.
2- It should be checked grammatically. For example, the parallel structure is not observed in the text.
Response:Some syntax is modified and highlighted in red.
3- The chemical composition of the raw material and mechanical properties such as yield stress, ultimate tensile strength, etc. are the results of tests performed by the authors? if yes, please describe the tests and report the results data such as the stress-strain diagram. if not, please refer to an appropriate reference.
Response:The chemical composition, yield stress, ultimate tensile strength, elastic modulus and Poisson's ratio, etc. were measured and given in Section 2. The engineering stress-strain diagram was added in Section 2.
4- How did you create the initial crack in various samples? Also, explain the author's method of ensuring the exact size of the pre-crack in all samples.
Response:First, fatigue cracks are prefabricated on the sample, the test temperature is room temperature, the loading method is axial loading, the stress ratio is 0.1, and the test frequency is 20Hz. The smaller cracks in the plastic zone are obtained by the method of step-by-step load reduction. The constant load control is adopted, and the load reduction amplitude of the adjacent level load does not exceed 20%, and the crack propagation length is about 2 or 3 mm. The detailed method is based on standard of ISO 12135: 2002. The standard was given in Section 2.
5- During the tests, what is the equipment used to measure cracks at different stages. please describe your method in full detail?
Response:The crack length is measured by the flexibility method. The detailed method is based on standard of ISO 12135: 2002.
6- Related to Figure 5, what is the measurement accuracy of crack length?
Response:According to the standard of ISO 12135: 2002, the measurement accuracy of crack length is 0.025 mm.
7- Related to Figure 7, the authors stated in the text that 10 samples were used to perform the test, But in this figure, there are nine diagrams as the test results. In addition, it is strongly suggested to re-draw all diagrams in one image for comparison easier than ago.
Response:In order to make the picture format neat, 9 curves were placed before. Now, all 10 curves are placed in Fig.9. All curves are added to one diagram in Fig.9(k).
We appreciate for Editors/Reviewers’ warm work earnestly, and hope that the correction will meet with approval.
Once again, thank you very much for your comments and suggestions.
Round 2
Reviewer 4 Report
The article has been improved after revision. I recommend to accept it in present form.
Author Response
Thanks very much for your kind work and consideration on publication of our paper. On behalf of my co-authors, we would like to express our great appreciation to editor and reviewers.